# A Low Body Mass Index Is Associated with Unsuccessful Treatment in Patients with *Mycobacterium avium* Complex Pulmonary Disease

**DOI:** 10.3390/jcm10081576

**Published:** 2021-04-08

**Authors:** Hironori Sadamatsu, Koichiro Takahashi, Hiroki Tashiro, Koji Kusaba, Tetsuro Haraguchi, Yuki Kurihara, Natsuko Komiya, Chiho Nakashima, Tomomi Nakamura, Shinya Kimura, Naoko Sueoka-Aragane

**Affiliations:** 1Division of Haematology, Respiratory Medicine and Oncology, Department of Internal Medicine, Faculty of Medicine, Saga University, Saga 849-8501, Japan; sadamatsu2hiro9@yahoo.co.jp (H.S.); si3222@cc.saga-u.ac.jp (H.T.); st8753@cc.saga-u.ac.jp (T.H.); chestn3yu.yu@gmail.com (Y.K.); natsu.komiya.438975@gmail.com (N.K.); d4208@cc.saga-u.ac.jp (C.N.); nakamurt@cc.saga-u.ac.jp (T.N.); shkimu@cc.saga-u.ac.jp (S.K.); sueokan@cc.saga-u.ac.jp (N.S.-A.); 2Department of Laboratory Medicine, Saga University Hospital, Saga 849-8501, Japan; kusabak@cc.saga-u.ac.jp

**Keywords:** *Mycobacterium avium* complex pulmonary disease, body mass index, cavitary lesions, prognostic nutritional index

## Abstract

Background: A low body mass index (BMI) has been reported to be a poor prognostic factor for *Mycobacterium avium* complex pulmonary disease (MAC-PD). The purpose of this study was to clarify the clinical features of MAC-PD in cases with a low BMI. Methods: This retrospective study analyzed the data of patients diagnosed with MAC-PD at Saga University Hospital between 2008 and 2019. The analyzed patient characteristics included age, gender, BMI, symptoms, laboratory data, chest computed tomography findings, and the treatment courses. We also investigated the factors associated with successful treatment. Results: In total, 144 patients were included in this study. The low-BMI group (BMI < 18.5 kg/m^2^) had a higher incidence of sputum, *Mycobacterium intracellurare* infection, and cavitary lesions, in addition to lower blood lymphocyte counts, higher neutrophil–lymphocyte ratios, and a lower prognostic nutritional index (PNI) when compared to the preserved-BMI group (BMI ≥ 18.5 kg/m^2^). Sixty-six of the 144 patients (45.8%) received treatment. Hemosputum, acid-fast bacillus sputum smear positivity, low lymphocyte counts, a low PNI, and unsuccessful treatment (48.5% vs. 24.2%, *p* < 0.05) were found to be associated with a low BMI. Conclusions: A low BMI is associated with cavitary lesions, malnutrition, and unsuccessful treatment in MAC-PD.

## 1. Introduction

The prevalence of nontuberculous mycobacteria pulmonary disease (NTM-PD) is increasing worldwide [1,2]. *Mycobacterium avium* complex (MAC), including *M. avium* and *M. intracellulare*, is the most common etiology of NTM-PD [3]. Despite the administration of multidrug combination therapy, including macrolides, disease progression and a poor prognosis are often observed in MAC pulmonary disease (MAC-PD). Older age, a low body mass index (BMI), low pulmonary function, anemia, malignancy, hemosputum, sputum smear positivity, and the presence of cavities have been reported to be poor prognostic factors of MAC-PD [4,5,6,7,8]. In particular, progressive cavity lesions are considered to be an important poor prognostic factor [9]. Of these poor prognostic factors, BMI is easily obtainable clinical information. The World Health Organization criteria define a BMI < 18.5 as underweight, 18.5 ≤ BMI < 25 as normal, 25 ≤ BMI < 30 as pre-obese, and BMI ≥ 30 as obese [10]. Therefore, a BMI < 18.5 kg/m^2^ has been regarded as a poor prognostic factor in previous studies [4,5].

A low BMI is considered to be a poor prognostic factor for MAC-PD, but the clinical features of low-BMI cases have not yet been fully elucidated. In the present study, we evaluated the clinical features of MAC-PD in cases with a low BMI.

## 2. Materials and Methods

### 2.1. Study Design and Patients

A total of 320 patients with a positive acid-fast bacillus (AFB) sputum culture and PD who visited Saga University Hospital between April 2008 and March 2019 were retrospectively analyzed. Patient data were collected from medical records. There were no human immunodeficiency virus-positive patients. Patients were diagnosed with pulmonary tuberculosis or NTM-PD, including MAC-PD, based on the culture of sputum or bronchial lavage fluid. In this study, patients who met the clinical and microbiologic criteria of the official American Thoracic Society/European Respiratory Society/European Society of Clinical Microbiology and Infectious Diseases/Infectious Diseases Society of America (ATS/ERS/ESCMID/IDSA) clinical practice guidelines were analyzed [11,12]. According to these criteria, nodules, cavitary lesions, and multifocal bronchiectasis with multiple small nodules on chest computed tomography (CT) were considered to be typical findings of MAC-PD. Additionally, MAC-PD was diagnosed for cases in which the AFB culture was positive for at least two sputum samples or at least one bronchial lavage fluid sample. Patient data included age, sex, BMI, smoking status, sputum smear findings, etiological organisms, underlying diseases, symptoms, laboratory data, chest CT findings, and treatment status.

### 2.2. Definition of Low BMI

BMI < 18.5 kg/m^2^ is defined as underweight in the World Health Organization criteria for obesity [10]. In this study, BMI < 18.5 kg/m^2^ was thus defined as low BMI, and BMI ≥ 18.5 kg/m^2^ was defined as preserved BMI.

### 2.3. Sputum Examination

Sputum cultures before treatment were performed using spontaneous sputum, induced sputum, and bronchial lavage fluid obtained at the time of diagnosis. Sputum cultures after treatment were performed using spontaneous sputum and induced sputum. Induced sputum was obtained by inhalation of 5% saline using a nebulizer.

### 2.4. Radiological Evaluation

Chest CT findings at the time of MAC-PD diagnosis were evaluated by two independent pulmonologists. The findings of representative patients are shown in Figure 1. Patients with multiple nodules and bronchiectasis were diagnosed with the nodular bronchiectatic (NB) form of MAC-PD. The NB form was further classified into non-cavitary NB (Figure 1A,B) or cavitary NB (Figure 1C) according to the absence or presence of cavitary lesions, respectively. Patients with fibrocavitary lesions mainly in the upper lobes were diagnosed with the fibrocavitary (FC) form of MAC-PD (Figure 1D). Patients with no specific pattern were diagnosed with an unclassified form of MAC-PD.

### 2.5. Definition of Laboratory Data

The neutrophil–lymphocyte ratio (NLR) and Onodera’s prognostic nutritional index (PNI) were used as nutritional indicators in this study. The NLR was initially reported as an indicator of the post-hospital admission course in patients admitted to intensive care units [13], and subsequently, it has been reported as a prognostic marker in malignant diseases [14,15]. The PNI was initially reported as an assessment tool of the perioperative nutritional status and surgical risk in patients with gastrointestinal surgery [16], and subsequently, it has also been reported as a prognostic marker in malignant diseases [17]. The NLR was calculated as the ratio of the neutrophil count to the lymphocyte count in peripheral blood [13]. The PNI score was calculated using the following formula at the time of MAC-PD diagnosis: 10 × serum albumin value (g/dL) + 0.005 × total lymphocyte count in the peripheral blood (/µL) [16,18]. Patients with a PNI score of less than 45 (cut-off value) were considered malnourished [17,19,20].

### 2.6. Definition of Treatment Outcomes

Treatment outcomes were assessed according to the NTM-NET consensus statement [21]. Sputum culture conversion was defined as the finding of at least three consecutive negative mycobacterial cultures, collected at least four weeks apart, during antimycobacterial treatment. The recurrence of sputum culture was defined as the re-emergence of at least two positive cultures with the causative species after cessation of antimycobacterial treatment. Conversion and recurrence were based on liquid media cultures. We defined cases with no sputum culture conversion or with a recurrence of sputum culture as refractory cases.

### 2.7. Ethical Approval

This study was approved by the ethics committee of Saga University Hospital (approval number: 2020-10-R-01; approval date: 28 December 2020), and was performed in accordance with the 1964 Declaration of Helsinki.

### 2.8. Statistical Analysis

All statistical analyses were performed using JMP Pro 14.2 (SAS Institute Inc., Cary, NC, USA). All data are presented as the mean and standard deviation. Patient groups were compared using the Mann–Whitney *U*-test and the χ^2^ test. Multivariate logistic regression analysis was performed with age, sex, and cavitary lesions as confounding factors to identify potential independent factors associated with a low BMI in the MAC-PD patients. For cavitary NB and FC, age and sex were used as confounding factors. *p*-values < 0.05 were considered to indicate statistical significance.

## 3. Results

### 3.1. Patient Selection

In total, the data from 320 patients with a positive AFB sputum culture and PD were collected (Figure 2). Among the 320 patients, 70 (21.9%) were diagnosed with pulmonary tuberculosis and 250 (78.1%) were diagnosed with NTM-PD. Among the patients with NTM-PD, 235 were diagnosed with MAC-PD; of these, 91 were excluded, because they did not satisfy the criteria of the official ATS/ERS/ESCMID/IDSA clinical practice guidelines or they had insufficient data for analysis. Finally, the data from 144 MAC-PD patients and 66 MAC-PD patients who received treatment were analyzed in this study.

### 3.2. Patient Characteristics

Patients were divided into a low-BMI (BMI <18.5 kg/m^2^) group and a preserved-BMI (BMI ≥18.5 kg/m^2^) group (Table 1). The mean BMI was 16.8 ± 1.3 kg/m^2^ in the low-BMI group and 21.4 ± 2.4 kg/m^2^ in the preserved-BMI group. AFB sputum smear positivity was more common in the low-BMI group than in the preserved-BMI group (67.7% vs. 50.0%, *p* < 0.05). *M. intracellulare* infection was significantly more common in the low-BMI group than in the preserved-BMI group (72.6% vs. 47.6%, *p* < 0.05). There was no difference in the rate of underlying diseases between the two groups, except for diabetes mellitus. Among the symptoms, sputum was significantly more common in the low-BMI group than in the preserved-BMI group (62.9% vs. 46.3%, *p* < 0.05). In the laboratory findings, the lymphocyte count was significantly lower (1381 ± 607/µL vs. 1580 ± 518/µL, *p* < 0.05) and the NLR was significantly higher (3.9 ± 2.6 vs. 3.1 ± 2.3, *p* < 0.05) in the low-BMI group than in the preserved-BMI group. There were significantly more patients with a PNI < 45 in the low-BMI group than in the preserved-BMI group (59.7% vs. 34.1%, *p* < 0.05). Among the chest CT findings, the prevalence of cavitary lesions, including cavitary NB and FC, was significantly higher in the low-BMI group than in the preserved-BMI group (35.5% vs. 20.7%, *p* < 0.05). Multivariate analysis was performed to identify potential independent factors associated with a low BMI in the MAC-PD patients (Table 2). *M. intracellulare* infection, the NLR, a PNI < 45, and the prevalence of cavitary lesions were positively correlated with a low BMI. The lymphocyte count and PNI score were negatively correlated with a low BMI.

### 3.3. Characteristics of the MAC-PD Patients Who Received Treatment

Among the 144 patients, 66 (45.8%) received treatment for MAC-PD (Table 3). The BMI was 16.6 ± 1.3 kg/m^2^ in the low-BMI group and 21.1 ± 2.0 kg/m^2^ in the preserved-BMI group. AFB sputum smear positivity before treatment was more common in the low-BMI group than in the preserved-BMI group (75.8% vs. 51.5%, *p* < 0.05). Among the symptoms, hemosputum was significantly more common in the low-BMI group than in the preserved-BMI group (33.3% vs. 9.1%, *p* < 0.05). In the laboratory findings, the lymphocyte count was significantly lower in the low-BMI group than in the preserved-BMI group (1367 ± 552/µL vs. 1661 ± 424/µL, *p* < 0.05). There were significantly more patients with a PNI < 45 in the low-BMI group than in the preserved-BMI group (60.6% vs. 33.3%, *p* < 0.05).

### 3.4. Treatment Outcomes

There was no difference between the low-BMI group and the preserved-BMI group in the administration rate of a standard regimen containing clarithromycin, rifampicin, and ethambutol, or in the use of aminoglycoside and fluoroquinolone (Table 4). The number of refractory cases with no sputum conversion was significantly higher in the low-BMI group than in the preserved-BMI group (48.5% vs. 24.2%, *p* < 0.05). Multivariate analysis was performed to identify potential independent factors associated with a low BMI in the MAC-PD patients who received treatment (Table 5). Hemosputum, a PNI < 45, no sputum conversion, and refractory cases were positively correlated with a low BMI. The lymphocyte count was negatively correlated with a low BMI.

### 3.5. Figures and Tables

Data from 320 patients with a positive AFB sputum culture and PD were collected. Among the 320 patients, 70 were diagnosed with pulmonary tuberculosis and 250 were diagnosed with NTM-PD. Among the patients with NTM-PD, 235 were diagnosed with MAC-PD; of these, 91 were excluded, because they did not satisfy the criteria of the official ATS/ERS/ESCMID/IDSA clinical practice guidelines or they had insufficient data for analysis. Finally, the data from 144 MAC-PD patients (Analysis 1) and 66 MAC-PD patients who received treatment (Analysis 2) were analyzed in this study. AFB, acid-fast bacillus; PD, pulmonary disease; NTM-PD, nontuberculous mycobacteria pulmonary disease; MAC-PD, *Mycobacterium avium* complex pulmonary disease.

## 4. Discussion

In the present study, a low BMI was associated with the presence of cavitary lesions, the detection of *M. intracellulare*, low blood lymphocyte counts, a high NLR, and a PNI < 45 in patients with MAC-PD. In addition, a low BMI was associated with hemosputum, low blood lymphocyte counts, a PNI < 45, and no sputum culture conversion in patients treated for MAC-PD.

The presence of cavitary lesions has been reported to be a poor prognostic factor in patients with MAC-PD. A previous study reported that among 634 MAC-PD patients, the 10-year mortality rate was 38.9% in patients with cavities and 9.0% in those without cavities [4]. Furthermore, among the NB forms, cavitary NB has been reported to have a worse prognosis than non-cavitary NB; among 782 MAC-PD patients, the 10-year mortality rate was 25.1% in patients with cavitary NB and 0.8% in those with non-cavitary NB [5]. Several studies have shown that BMI is lower in patients with cavities than in those without cavities [5,22]. In the present study, the prevalence of cavitary lesions was significantly higher in the low-BMI group than in the preserved-BMI group. Therefore, low BMI and cavitary lesions appear to be closely related in patients with MAC-PD.

Malnutrition is often seen in patients with MAC-PD. The usefulness of serum albumin levels, total cholesterol levels, and blood lymphocyte counts as markers for the screening of malnutrition has been shown [23]. The PNI and NLR have also been reported to be useful as prognostic markers in malignant diseases [14,15,17,24]. Onodera’s PNI is a useful indicator for malnutrition that can be calculated from only the serum albumin level and blood lymphocyte count [16]. A previous study showed that MAC-PD patients with a PNI < 45 had a higher mortality rate, a higher prevalence of cavitary lesions, and a lower BMI than MAC-PD patients with a PNI ≥ 45 [25]. In the present study also, there were significantly more patients with a PNI < 45 in the low-BMI group than in the preserved-BMI group. Moreover, a PNI < 45 was associated with a low BMI in the multivariate analysis. Thus, in addition to a PNI < 45, a low BMI may be useful as an indicator of malnutrition and a poor prognosis in patients with MAC-PD.

The NLR is a nutritional evaluation tool that uses the levels of blood cell components, and it is applied as a prognostic marker [14,15]. The NLR is higher in pulmonary tuberculosis patients with cavities than in those without cavities [26]. In the present study, the NLR was significantly higher in the low-BMI group than in the preserved-BMI group. The NLR was associated with a low BMI in the multivariate analysis. Thus, a low BMI, PNI, and NLR may be useful indicators of cavitary lesions in patients with MAC-PD.

Along with older age, low pulmonary function, hemosputum, AFB sputum smear positivity, and the presence of cavitary lesions, a low BMI is considered to be a poor prognostic factor of MAC-PD [4,5,6,7,8]. A previous study showed that the use of a standard regimen containing clarithromycin, rifampicin, and ethambutol is associated with sputum conversion and reduced disease progression [27]. In the present study, there were significantly more patients with no sputum conversion in the low-BMI group than in the preserved-BMI group, even though there was no difference in the administration rate of the standard regimen between the two groups. Additionally, no sputum conversion was associated with a low BMI in the multivariate analysis. Hence, a low BMI may be associated with a poor response to treatment.

In the present study, a low BMI was associated with cavitary lesions, the PNI, the NLR, and refractory disease. The PNI and NLR are associated not only with the nutritional status, but also with the immune status, because blood cell components are used to determine the PNI and NLR. A low BMI may be associated with refractory disease through malnutrition and compromised immunity. Furthermore, the BMI was significantly lower in refractory cases (*n* = 24) than in non-refractory cases (*n* = 42), although the presence of cavitary lesions, the PNI, and the NLR did not differ significantly between the two groups in this study (data not shown). A low BMI may contribute more to unsuccessful treatment than other factors.

The present results must be considered in light of some limitations. First, since the study was retrospective in nature, the treatment periods and therapeutic drugs used were not unified. Second, the post-treatment follow-up periods for MAC-PD differed among the patients. Finally, the present study involved patients at a single center, and the sample size was small; to confirm the validity of the present results, multi-center prospective studies with larger numbers of patients should be performed.

## 5. Conclusions

A low BMI was associated with other poor prognostic factors of MAC-PD, including the presence of cavitary lesions and hemosputum in patients with MAC-PD. A low BMI was also associated with the detection of *M. intracellulare*, a low blood lymphocyte count, a high NLR, a PNI < 45, and no sputum conversion. Thus, low BMI may be a convenient and useful predictor of a poor prognosis in MAC-PD.

## Figures and Tables

**Figure 1 jcm-10-01576-f001:**
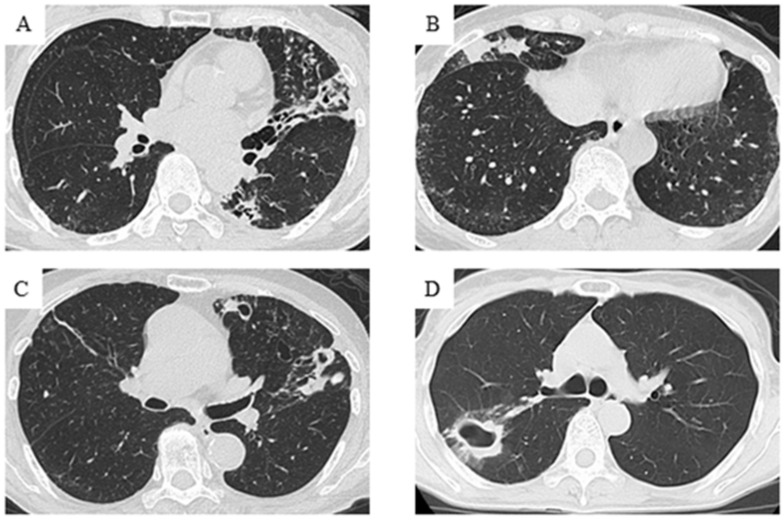
Chest computed tomography findings of representative patients. (**A**–**C**) Patients with multiple nodules and bronchiectasis were diagnosed with the NB form of MAC-PD. The NB form was further classified into non-cavitary NB (**A**,**B**) or cavitary NB (**C**) according to the absence or presence of cavitary lesions, respectively. (**D**) Patients with fibrocavitary lesions mainly in the upper lobes were diagnosed with the FC form of MAC-PD. NB, nodular bronchiectatic; MAC-PD, *Mycobacterium avium* complex pulmonary disease; FC, fibrocavitary.

**Figure 2 jcm-10-01576-f002:**
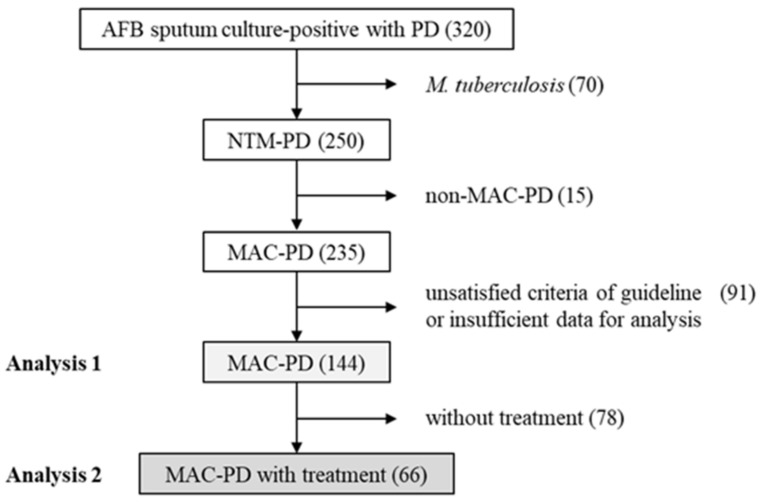
Flowchart of the study. NTM-PD, nontuberculous mycobacteria pulmonary disease; PD, pulmonary disease.

**Table 1 jcm-10-01576-t001:** Clinical characteristics of the MAC-PD patients with a BMI < 18.5 kg/m^2^ or ≥18.5 kg/m^2^.

	Total	BMI < 18.5	BMI ≥ 18.5	*p*-Value
Subjects, *n* (%)	144	62 (43.1)	82 (56.9)	
Age (years)	70.9 ± 10.0	70.8 ± 10.1	70.8 ± 10.0	0.64
Sex (male/female)	36/108	13/49	23/59	0.33
BMI (kg/m^2^)	19.4 ± 3.0	16.8 ± 1.3	21.4 ± 2.4	
Non-smokers	94 (75.8)	38 (76.0)	56 (75.7)	0.97
Sputum smear-positive	83 (57.6)	42 (67.7)	41 (50.0)	<0.05
**Etiological organism**				
*M. avium*	64 (44.4)	19 (30.6)	45 (54.9)	<0.05
*M. intracellulare*	84 (58.3)	45 (72.6)	39 (47.6)	<0.05
**Underlying disease**				
History of tuberculosis	11 (7.6)	6 (9.7)	5 (6.1)	0.42
COPD	5 (3.5)	2 (3.2)	3 (3.7)	0.89
Diabetes mellitus	19 (13.2)	4 (6.5)	15 (18.3)	<0.05
Connective tissue diseases	29 (20.1)	16 (25.8)	13 (15.9)	0.14
Malignancy	27 (18.8)	12 (19.4)	15 (18.3)	0.87
Chemotherapy for malignancy	3 (2.1)	2 (3.2)	1 (1.2)	0.40
Corticosteroid use	20 (13.9)	7 (11.3)	13 (15.9)	0.43
Immunosuppressive drug use *	2 (1.4)	1 (1.6)	1 (1.2)	0.84
**Symptoms**				
Fever (>37.5 °C)	11 (7.6)	6 (9.7)	5 (6.1)	0.42
Cough	90 (62.5)	42 (67.7)	48 (58.5)	0.26
Sputum	77 (53.5)	39 (62.9)	38 (46.3)	<0.05
Hemosputum	24 (16.7)	12 (19.4)	12 (14.6)	0.45
**Laboratory data**				
WBC (/µL)	6297 ± 1864	6334 ± 1863	6268 ± 1875	0.99
Neutrophil (/µL)	4244 ± 1782	4381 ± 1855	4141 ± 1731	0.42
Lymphocyte (/µL)	1494 ± 565	1381 ± 607	1580 ± 518	<0.05
NLR	3.4 ± 2.5	3.9 ± 2.6	3.1 ± 2.3	<0.05
Albumin (g/dL)	3.6 ± 0.5	3.6 ± 0.5	3.8 ± 0.5	0.10
PNI score	44.5 ± 7.0	43.2 ± 7.2	45.5 ± 6.7	<0.05
PNI < 45	65 (45.1)	37 (59.7)	28 (34.1)	<0.05
CRP (mg/dL)	0.8 ± 1.5	1.0 ± 1.7	0.6 ± 1.4	0.11
**Chest CT findings**				
Non-cavitary NB	102 (70.8)	40 (64.5)	62 (75.6)	0.15
Cavitary NB	33 (22.9)	18 (29.0)	15 (18.3)	0.13
FC	6 (4.2)	4 (6.5)	2 (2.4)	0.23
Unclassified	3 (2.1)	0 (0.0)	3 (3.7)	0.13
Cavitary NB+FC	39 (27.1)	22 (35.5)	17 (20.7)	<0.05
With treatment	66 (45.8)	33 (53.2)	33 (40.2)	0.12

* Two patients took tacrolimus for connective tissue diseases. MAC-PD, *Mycobacterium avium* complex pulmonary disease; BMI, body mass index; COPD, chronic obstructive pulmonary disease; WBC, white blood cell; NLR, neutrophil–lymphocyte ratio; PNI, prognostic nutritional index; CRP, c-reactive protein; CT, computed tomography; NB, nodular bronchiectatic; FC, fibrocavitary. Data are presented as mean ± standard deviation or *n* (%).

**Table 2 jcm-10-01576-t002:** Multivariate analysis of the MAC-PD patients with a BMI < 18.5 kg/m^2^.

	Multivariate Analysis		
	OR or β	95%CI	*p*-Value
Sputum smear-positive	1.94	0.97–3.90	0.06
**Etiological organism**			
*M. avium*	0.37	0.17–0.76	<0.05
*M. intracellulare*	2.82	1.34–6.13	<0.05
**Underlying disease**			
Diabetes mellitus	0.34	0.09–1.05	0.06
**Symptoms**			
Sputum	1.82	0.92–3.64	0.09
**Laboratory data**			
Lymphocyte (/µL)	0.13	0.02–0.88	<0.05
NLR	9.35	1.16–98.94	<0.05
PNI score	0.16	0.03–0.89	<0.05
PNI < 45	3.14	1.50–6.80	<0.05
**Chest CT findings**			
Cavitary NB+FC	2.15	1.02–4.63	<0.05

MAC-PD, *Mycobacterium avium* complex pulmonary disease; BMI, body mass index; NLR, neutrophil-lymphocyte ratio; PNI, prognostic nutritional index; CT, computed tomography; NB, nodular bronchiectatic; FC, fibrocavitary; OR, odds ratio; CI, confidence interval.

**Table 3 jcm-10-01576-t003:** Clinical characteristics of the MAC-PD patients with a BMI < 18.5 kg/m^2^ or ≥18.5 kg/m^2^ who received treatment.

	Total	BMI < 18.5	BMI ≥ 18.5	*p*-Value
Subjects, *n* (%)	66	33 (50.0)	33 (50.0)	
Age (years)	69.4 ± 10.7	69.0 ± 9.7	69.8 ± 11.8	0.64
Sex (male/female)	17/49	7/26	10/23	0.40
BMI (kg/m^2^)	18.8 ± 2.8	16.6 ± 1.3	21.1 ± 2.0	
Non-smokers	44 (75.9)	22 (75.9)	22 (75.9)	1.00
Sputum smear-positive	42 (63.6)	25 (75.8)	17 (51.5)	<0.05
**Etiological organism**				
*M. avium*	28 (42.4)	12 (36.4)	26 (48.5)	0.32
*M. intracellulare*	40 (60.6)	23 (69.7)	17 (51.5)	0.13
**Underlying disease**				
History of tuberculosis	7 (10.6)	4 (12.1)	3 (9.1)	0.69
COPD	3 (4.6)	1 (3.0)	2 (6.1)	0.55
Diabetes mellitus	9 (13.6)	4 (12.1)	5 (15.1)	0.76
Connective tissue diseases	14 (21.2)	8 (24.2)	6 (18.2)	0.55
Malignancy	8 (12.1)	3 (9.1)	5 (15.2)	0.45
Chemotherapy for malignancy	0 (0.0)	0 (0.0)	0 (0.0)	
Corticosteroid use	7 (10.6)	2 (6.1)	5 (15.2)	0.23
Immunosuppressive drug use *	1 (1.5)	1 (3.0)	0 (0.0)	0.31
**Symptoms**				
Fever (>37.5 °C)	10 (15.2)	5 (15.2)	5 (15.2)	1.00
Cough	48 (72.7)	26 (78.8)	22 (66.7)	0.27
Sputum	41 (62.1)	24 (72.7)	17 (51.5)	0.08
Hemosputum	14 (21.2)	11 (33.3)	3 (9.1)	<0.05
**Laboratory data**				
WBC (/µL)	6417 ± 2042	6333 ± 1966	6500 ± 2142	0.79
Neutrophil (/µL)	4253 ± 1976	4288 ± 2040	4217 ± 1941	0.88
Lymphocyte (/µL)	1516 ± 510	1367 ± 552	1661 ± 424	<0.05
NLR	3.3 ± 2.4	3.9 ± 2.9	2.8 ± 1.6	0.12
Albumin (g/dL)	3.7 ± 0.6	3.6 ± 0.5	3.7 ± 0.6	0.53
PNI score	44.2 ± 6.9	43.1 ± 7.2	45.3 ± 6.5	0.11
PNI < 45	31 (47.0)	20 (60.6)	11 (33.3)	<0.05
CRP (mg/dL)	1.0 ± 1.8	1.1 ± 1.5	0.9 ± 2.0	0.69
**Chest CT findings**				
Non-cavitary NB	33 (50.0)	14 (42.4)	19 (57.6)	0.22
Cavitary NB	27 (40.9)	16 (48.5)	11 (33.3)	0.21
FC	5 (7.6)	3 (9.1)	2 (6.1)	0.64
Unclassified	1 (1.5)	0 (0.0)	1 (3.0)	0.31
Cavitary NB+FC	32 (48.5)	19 (57.6)	13 (39.4)	0.14

* A patient took tacrolimus for connective tissue diseases. MAC-PD, *Mycobacterium avium* complex pulmonary disease; BMI, body mass index; COPD, chronic obstructive pulmonary disease; WBC, white blood cell; NLR, neutrophil–lymphocyte ratio; PNI, prognostic nutritional index; CRP, c-reactive protein; CT, computed tomography; NB, nodular bronchiectatic; FC, fibrocavitary. Data are presented as mean ± standard deviation or *n* (%).

**Table 4 jcm-10-01576-t004:** Treatment regimens and outcomes of the MAC-PD patients with a BMI < 18.5 kg/m^2^ or ≥18.5 kg/m^2^.

	Total	BMI < 18.5	BMI ≥ 18.5	*p*-Value
Subjects, *n* (%)	66	33 (50.0)	33 (50.0)	
**Treatment content**				
Treatment duration (months)	18.6 ± 12.2	21.1 ± 15.6	15.7 ± 5.5	0.18
CAM+RFP+EB regimen	42 (63.6)	24 (72.7)	18 (54.5)	0.12
Aminoglycoside use	10 (15.2)	5 (15.2)	5 (15.2)	1.00
Fluoroquinolone use	13 (19.7)	7 (21.2)	6 (18.2)	0.76
**After treatment**				
CAM resistance	9 (13.6)	4 (12.1)	5 (15.2)	0.72
No sputum conversion	10 (15.2)	8 (24.2)	2 (6.1)	<0.05
Recurrence sputum culture	14 (21.2)	8 (24.2)	6 (18.2)	0.55
Refractory cases	24 (36.4)	16 (48.5)	8 (24.2)	<0.05

MAC-PD, *Mycobacterium avium* complex pulmonary disease; BMI, body mass index; CAM, clarithromycin; RFP, rifampicin; EB, ethambutol. CAM resistance was defined as minimum inhibitory concentration > 32 µg/mL. Refractory cases were defined as cases with no sputum culture conversion or a recurrence of sputum culture. Data are presented as mean ± standard deviation or *n* (%).

**Table 5 jcm-10-01576-t005:** Multivariate analysis of the MAC-PD patients with a BMI < 18.5 kg/m^2^ who received treatment.

	Multivariate Analysis		
	OR or β	95%CI	*p*-Value
Sputum smear-positive	2.84	0.99–8.69	0.05
**Symptoms**			
Hemosputum	5.39	1.36–27.92	<0.05
**Laboratory data**			
Lymphocyte (/µL)	0.05	0.00–0.63	<0.05
PNI < 45	4.59	1.42–16.53	<0.05
**After treatment**			
No sputum conversion	5.56	1.18–41.34	<0.05
Refractory cases	2.94	1.02–9.06	<0.05

MAC-PD, *Mycobacterium avium* complex pulmonary disease; BMI, body mass index; PNI, prognostic nutritional index; OR, odds ratio; CI, confidence interval.

## Data Availability

The data are contained within the article.

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
