# Peer review of "A Low Body Mass Index Is Associated with Unsuccessful Treatment in Patients with Mycobacterium avium Complex Pulmonary Disease"

_jcm, 2021, doi:10.3390/jcm10081576_

Round 1

Reviewer 1 Report

The authors present a retrospective review that shows that low BMI is associated with poor treatment response in MAC pulmonary disease.

I would appreciate inclusion of hemoglobin and Mean Corpuscular Volume in the lab indices. Others have shown correlation of anemia with severity of MAC-PD. If anemia is present is it a consequence of iron deficiency or anemia of chronic disease? In my own experience correction of iron deficiency anemia has improved immune response in advanced chronic MAC pulmonary disease.

Author Response

Dear Dr. reviewer 1

Date: April 1st, 2021

Thank you for kindly suggestion to our manuscript.

 We wish to express our appreciation to the reviewer for his or her insightful comments, which have helped us significantly improve the paper. The replies to the reviewer’s comments point-by-point are shown the attached file.

We revised articles entitled, “A low body mass index is associated with unsuccessful treatment in patients with Mycobacterium avium complex pulmonary disease” to be reviewed for consideration for publication in Journal of Clinical Medicine.

Sincerely,

Koichiro Takahashi MD, PhD

Division of Hematology, Respiratory Medicine and Oncology, Department of Internal Medicine, Faculty of Medicine, Saga University

5-1-1 Nabeshima, Saga 849-8501, Japan

Tel: +81-952-34-2372, Fax: +81-952-34-2017

Reviewer 2 Report

This cohort study looks at factors associated with poor outcomes including failure to sputum convert, and focusses on BMI. It is an impressively large cohort for an NTM study in a single centre, and has a good amount of data on every patient. The English is clear, structure of the paper flows well and data presentation is appropriate. There are some matters that could be improved

  1. Methods - the statistics are quite basic in the sense that only univariate analyses are done comparing high and low BMI. Whilst some differences in clinical baseline parameters (eg connective tissue disease) are not statistically different between groups they are potential confounders and a plan to adjust for these in comparisons of outcomes should be added to methods and followed through to results. A key one that did appear different was cavitatary disease +/- fibrocavitatary change - this could be cause or effect of low BMI and is thus particularly relevant to control for when thinking of outcomes. Multivariable analyses taking key covariates into account and justifying the choice of them could be added. Detail on how much immunosuppression was used might be helpful (steroids and chemo are mentioned but not others)
  2. Results - addition of multivariable analyses as above. This might be possible to derive a prognostic score using the various variables discussed as relevant to prognosis, but depends on the model

Author Response

Dear Dr. reviewer 2

Date: April 1st, 2021

Thank you for kindly suggestion to our manuscript.

 We wish to express our appreciation to the reviewer for his or her insightful comments, which have helped us significantly improve the paper. The replies to the reviewer’s comments point-by-point are shown the attached file.

We revised articles entitled, “A low body mass index is associated with unsuccessful treatment in patients with Mycobacterium avium complex pulmonary disease” to be reviewed for consideration for publication in Journal of Clinical Medicine.

Sincerely,

Koichiro Takahashi MD, PhD

Division of Hematology, Respiratory Medicine and Oncology, Department of Internal Medicine, Faculty of Medicine, Saga University

5-1-1 Nabeshima, Saga 849-8501, Japan

Tel: +81-952-34-2372, Fax: +81-952-34-2017

Reviewer 3 Report

In this study, Sadamatsu et al. reported that low BMI was associated with other poor prognostic factors of MAC-PD. It is potentially interesting study that suggests low BMI would be one of the cause of progressive disease type. However, deeper analysis of causal relationships between each factors are necessary.

Low BMI is well established marker for poor prognosis, also, cavity and Mycobacterium intracellurare infection are associated with poor prognosis (Jhun, BW et al. Eur Respir J 2020;55(1)., doi:10.5588/ijtld.11.0534 (2012)., Fukushima, K et al.  Sci. Rep. 2021, 11, 1178, doi:10.1038/s41598-021-81025-w.).

Hence, attribution of causality of prognosis by one of these factors is challenging, but important. For example, progressive cavitary disease may be the cause of low BMI and malnutrition, or vice versa.

Major comments

  • Many confounding factors would affected the results that “low BMI is associated with cavitary lesions, malnutrition, and unsuccessful treatment”.

Strict statistical analysis such as multivariate regression or propensity score matching analysis would be needed to obtain more persuasive conclusion.

  • Lines 51-52 “A total of 320 patients with a positive acid-fast bacillus (AFB) sputum smear and PD who visited Saga University Hospital between 2008 and 2019 were retrospectively analyzed.” Inclusion of only AFB positive patients would lead to selection bias. Also, what is the definition of PD? Authors should describe more precise definition.

  • Lines 56-57 “MAC-PD was diagnosed for cases in which AFB culture was positive at least once from sputum or bronchial lavage fluid, and chest computed tomography (CT) showed typical findings of MAC-PD.” This diagnostic criteria for MAC-PD is not standard. Therefore, it reduces the applicability of their results.

  • Hence, authors should review all patients or all acid fast bacilli culture positive patients. And, enroll MAC-PD patients who satisfied the current guidelines for the diagnosis of MAC-PD (Griffith, D. E. et al. An official ATS/IDSA statement: doi:10.1164/rccm.200604-571ST (2007), Daley, C. L. et al. An Official ATS/ERS/ESCMID/IDSA Clinical Practice Guideline, doi:10.1093/cid/ciaa241 (2020).).

  • Line 143-145 “We defined cases with no sputum culture conversion or a recurrent sputum culture after treatment as refractory cases.” This should be moved to methods section. It is difficult to interpret culture conversion without knowing if specimens were collected in a standardized fashion and how many were obtained both before and after conversion. More details about this and whether conversion was based on solid or liquid media would be helpful. Use of standardized outcome measures would facilitate comparison with more recent studies (van Ingen, J. et al. The European respiratory journal 51, doi:10.1183/13993003.00170-2018 (2018).).

Minor comments

Line 52 “between 2008 and 2019”. What month?

Author Response

Dear Dr. reviewer 3

Date: April 1st, 2021

Thank you for kindly suggestion to our manuscript.

 We wish to express our appreciation to the reviewer for his or her insightful comments, which have helped us significantly improve the paper. The replies to the reviewer’s comments point-by-point are shown the attached file.

We revised articles entitled, “A low body mass index is associated with unsuccessful treatment in patients with Mycobacterium avium complex pulmonary disease” to be reviewed for consideration for publication in Journal of Clinical Medicine.

Sincerely,

Koichiro Takahashi MD, PhD

Division of Hematology, Respiratory Medicine and Oncology, Department of Internal Medicine, Faculty of Medicine, Saga University

5-1-1 Nabeshima, Saga 849-8501, Japan

Tel: +81-952-34-2372, Fax: +81-952-34-2017

Round 2

Reviewer 3 Report

Authors fully addressed my concerns.